# How SGD Selects the Global Minima in Over-parameterized Learning: A Dynamical Stability Perspective

**Lei Wu**
School of Mathematical Sciences
Peking University
Beijing, 100081, P.R. China
`leiwu@pku.edu.cn`

**Chao Ma**
Program in Applied and Computational Mathematics
Princeton University
Princeton, NJ 08544, USA
`chaom@princeton.edu`

**Weinan E**
Department of Mathematics and Program in Applied and Computational Mathematics
Princeton University, Princeton, NJ 08544, USA and
Beijing Institute of Big Data Research, Beijing, 100081, P.R. China
`weinan@math.princeton.edu`

## Abstract

The question of which global minima are accessible by a stochastic gradient decent (SGD) algorithm with specific learning rate and batch size is studied from the perspective of dynamical stability. The concept of non-uniformity is introduced, which, together with sharpness, characterizes the stability property of a global minimum and hence the accessibility of a particular SGD algorithm to that global minimum. In particular, this analysis shows that learning rate and batch size play different roles in minima selection. Extensive empirical results seem to correlate well with the theoretical findings and provide further support to these claims.

## 1 Introduction

In machine learning we have always faced with the following dilemma: The function that we minimize is the empirical risk but the one that we are really interested in is the population risk. In the old days when typical models have only few isolated minima, this issue was not so pressing. But now in the setting of over-parametrized learning, e.g. deep learning, when there is a large set of global minima, all of which have zero training error but the test error can be very different, this issue becomes highly relevant. In fact one might say that the task of optimization algorithms has become: Find the set of parameters with the smallest test error among all the ones with zero training error.

At the moment, this is clearly an impossible task since we do not have much explicit information about the population risk. Therefore in this paper, we take a limited view and ask the question: Which global minima (of the empirical risk, of course) are accessible to a particular optimization algorithm with a particular set of hyper-parameters? In other words, how the different optimization algorithms with different hyper-parameters pick out the different set of global minima?

Specifically, in deep learning one of the most puzzling issues is the recent observation that SGD tends to select the so-called flat minima, and flatter minima seem to generalize better [3, 13, 7]. Several very interesting attempts have been made to understand this issue. Goyal et al. [2] and Hoffer et al. [4] numerically studied how the learning rate and batch size impact test accuracy of the solutions found by SGD. Jastrzębski et al. [6] suggested that the ratio between the learning rate and the batch

size $\eta/B$ is a key factor that affects the flatness of minima selected. Zhu et al. [15] demonstrated that the specific non-isotropic structure of the noise is important for SGD to find flat minima. Of particular interest to this work is the observation in [15] that the minima found by GD (gradient decent) can be unstable for SGD. As shown in Figure 1, when switching the algorithm from GD to SGD at a point close to a global minimum, SGD escapes from that minimum and converges to another global minimum which generalizes better. The time it takes for the escape is very short compared to that required for SGD to converge.

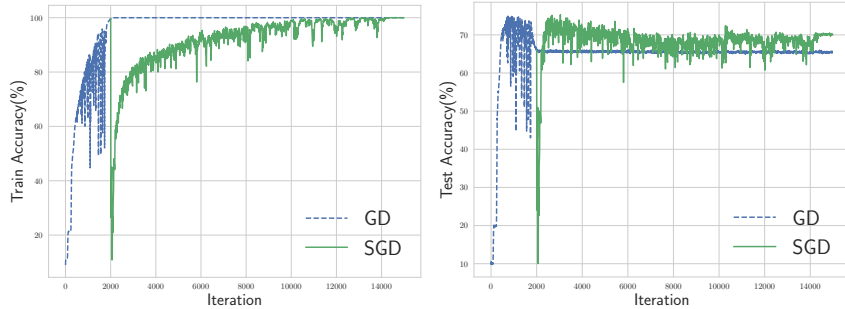

Figure 1: Fast escape phenomenon in fitting corrupted FashionMNIST. When the optimizer is switched from GD to SGD with the same learning rate, though the GD iterations is already quite close to a global minimum, one observes a fast escape from that global minimum and subsequent convergence to another global minimum. As shown by the right figure, the ultimate global minimum found by SGD generalizes better for this example.

In this paper, we make an attempt to study these issues systematically from a dynamical stability perspective. Our focus will be on SGD and GD. But the principle is applicable to any other optimization algorithms. We begin our analysis by explaining the escape phenomenon in Figure 1 using a toy example that can reproduce the basic features of this process. We then formalize the intuition into an analytically argument. The analysis leads to a sharpness-non-uniformity diagram that characterizes the kind of minima that are stable and hence accessible for SGD. We show that both the sharpness and the non-uniformity are important for the selection of the global minima, although we do observe in experiments that these two quantities are strongly correlated for deep neural network models. Our extensive numerical results give strong support to the theoretical analysis.

## 2    The Mechanism of Escape

To understand how the fast escape happens, let us first consider a one-dimensional problem $f(x) = \frac{1}{2}\left(f_1(x) + f_2(x)\right)$ with

$$f_1(x) = \min\{x^2, 0.1(x-1)^2\}, \ f_2(x) = \min\{x^2, 1.9(x-1)^2\}.$$

The landscape is shown in Figure 2. This function has two global minima at $x = 0$ and $x = 1$. We find that for $x_0 = 1 + \varepsilon$, SGD always escapes and converges to $x = 0$ instead of staying in the initial basin, as long as $\varepsilon \neq 0$ and $\eta$ is relatively large (one trajectory of is shown in Figure 2 in red). In contrast, SGD starting from $x_0 = \varepsilon$ with the same learning rate always converges to $x = 0$, and we never observe the escape phenomenon. As a comparison, we observe that GD starting from the neighborhood of the two minima with the same learning rate will behave the same way.

**Intuitive explanation**    For this simple case, the above phenomena can be easily explained. (1) The two minima has the same sharpness $f'' = 1$ , so when the learning rate is small enough ($\eta \leq \frac{2}{f''} = 2$ in this problem), the two minima are both stable for GD. (2) However, for SGD, in each iteration, it randomly picks one function from $f_1$ and $f_2$ and applies gradient descent to that function. Since $f_1''(1) = 0.1, f_2''(1) = 1.9$, SGD with the learning rate $\eta = 0.7$ is stable for $f_1$ but unstable for $f_2$. Thus $x = 1$ is not stable. In contrast, $\eta = 0.7$ is stable for both $f_1$ and $f_2$ around $x = 0$ since $f_1''(0) = f_2''(0) = 1$. This intuition can be formalized by the following stability analysis.

**Formal argument**    Without loss of generality, let us assume that $x = 0$ is the global minimum of interest. Consider the following more general one-dimensional optimization problem, $f(x) =$

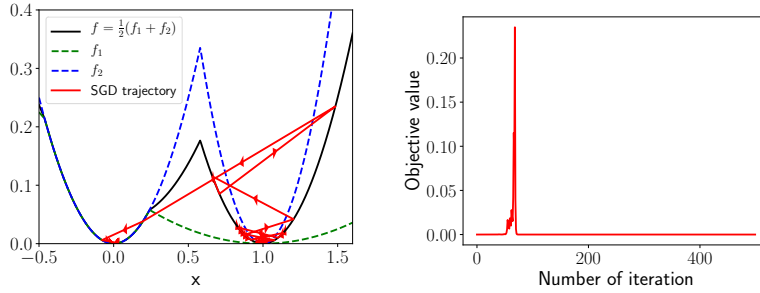

Figure 2: Motivating example. (Left) One trajectory of SGD with learning rate $\eta = 0.7, x_0 = 1 - 10^{-5}$, showing convergence to 0. GD with the same learning rate will converge to 1. (Right) The value of objective function, showing a burst during the escape.

$\frac{1}{2n} \sum_{i=1}^{n} a_i x^2$ with $a_i \geq 0 \ \forall i \in [n]$. Thus the SGD with batch size $B = 1$ is given by:

$$x_{t+1} = x_t - \eta a_\xi x_t = (1 - \eta a_\xi) x_t, \tag{1}$$

where $\xi$ is a random variable that satisfies $\mathbb{P}(\xi = i) = 1/n$. Hence, we have

$$\mathbb{E}x_{t+1} = (1 - \eta a)^t x_0, \tag{2}$$

$$\mathbb{E}x_{t+1}^2 = \left[(1 - \eta a)^2 + \eta^2 s^2\right]^t x_0^2, \tag{3}$$

where $a = \sum_{i=1}^{n} a_i/n$, $s = \sqrt{\sum_{i=1}^{n} a_i^2/n - a^2}$. Therefore, for SGD to be stable around $x = 0$, we not only need $|1 - \eta a| \leq 1$ (stability condition of GD), but also $(1 - \eta a)^2 + \eta^2 s^2 \leq 1$. Otherwise, $x_t$ will blow up exponentially fast. In particular, SGD can only select the minima with $s \leq 1/\eta$ whereas GD does not have this requirement.

In particular, for the toy example discussed above, at $x = 1$, we have $s = 1.8 > 1/0.7$, so as predicted SGD with $\eta = 0.7$ will escape and finally converge to minimum $x = 0$, where $s = 0 < 1/0.7$.

## 3 Linear Stability Analysis

Let us consider the minimization of the training error

$$f(x) = \frac{1}{n} \sum_{i=1}^{n} f_i(x)$$

by a general optimizer

$$x_{t+1} = x_t - G(x_t; \xi_t) \tag{4}$$

where $\xi_t$ is a random variable independent of $x_t$, and $\xi_t$ are *i.i.d.*. For SGD, $G(x_t; \xi_t) = \eta \nabla f_{\xi_t}(x_t)$; for GD, $G(x_t; \xi_t) = \eta \sum_{i=1}^{n} \nabla f_i(x_t)/n$. In practice, $G(x_t; \xi_t)$ usually depends on some tunable hyper-parameters, like learning rate, batch size, etc..

**Definition 1** (Fixed point). We say $x^*$ is a fixed point of stochastic dynamics (4), if for any $\xi$, we have $G(x^*; \xi) = 0$.

It should be remarked that this kind of fixed points do not always exists. However for the over-parametrized learning (OPL) problems of interest, all the global minima of $f(x)$ are fixed points of the popular optimizers such as SGD, Adam, etc.. Note that for a specific optimizer, only the dynamically stable fixed points can be selected. If a fixed point is unstable, a small perturbation will drive the optimizer to move away. To formalize this, we introduce a kind of stability concept for the stochastic dynamics (4), which is an extension of the classical notion of linear stability in dynamical systems [11].

**Definition 2** (Linear stability). Let $x^*$ be a fixed point of stochastic dynamics (4). Consider the linearized dynamical system:

$$\tilde{x}_{t+1} = \tilde{x}_t - A_{\xi_t}(\tilde{x}_t - x^*), \tag{5}$$

where $A_{\xi_t} = \nabla_x G(x^*, \xi_t)$. We say that $x^*$ is *linearly stable* if there exists a constant $C$ such that,

$$\mathbb{E}[\|\tilde{x}_t\|^2] \leq C\|\tilde{x}_0\|^2, \text{ for all } t > 0. \tag{6}$$

### 3.1 Stochastic Gradient Descent

In this section, we derive the stability condition for SGD. Let $x^*$ be the fixed point of interest. Consider the quadratic approximation of $f$ near $x^*$: $f(x) \approx \frac{1}{2n} \sum_{i=1}^{n} (x - x^*)^\top H_i (x - x^*)$ with $H_i = \nabla^2 f_i(x^*)$. Here we have assumed $f(x^*) = 0$. The corresponding linearized SGD is given by

$$x_{t+1} = x_t - \frac{\eta}{B} \sum_{j=1}^{B} H_{\xi_j}(x_t - x^*) \tag{7}$$

where $B$ is the batch size and $\xi = \{\xi_1, \cdots, \xi_B\}$ is a uniform, non-replaceable random sampling of size $B$ on $\{1, 2, \cdots, n\}$. To characterize the stability of this dynamical system, we need the following two quantities.

**Definition 3.** Let $H = \frac{1}{n} \sum_{i=1}^{n} H_i, \Sigma = \frac{1}{n} \sum_{i=1}^{n} H_i^2 - H^2$. We define $a = \lambda_{\max}(H)$ to be the *sharpness*, and $s = \lambda_{\max}(\Sigma^{1/2})$ to be the *non-uniformity*, respectively.

**Theorem 1.** *The global minimum $x^*$ is linearly stable for SGD with learning rate $\eta$ and batch size $B$ if the following condition is satisfied*

$$\lambda_{\max}\left\{ (I - \eta H)^2 + \frac{\eta^2(n - B)}{B(n-1)} \Sigma \right\} \leq 1. \tag{8}$$

*When $d = 1$, this becomes a sufficient and necessary condition.*

The proof can be found in Appendix A

When $d = 1$, this becomes the condition $(1 - \eta a)^2 + \eta^2 s^2 \leq 1$ introduced in Section 2. The condition (8) is sharp but not intuitive. A less sharp but simpler necessary condition for guaranteeing (8) is as follows

$$0 \leq a \leq \frac{2}{\eta}, \quad 0 \leq s \leq \frac{1}{\eta}\sqrt{\frac{B(n-1)}{n-B}}. \tag{9}$$

This is obtained by requiring the two terms at the left hand side of (8) to satisfy the stability condition separately. In particular, when $B \ll n$ the largest non-uniformity allowed is roughly $\sqrt{B}/\eta$. As shown in the next section, numerical experiments show that in deep learning the condition (9) is quite sharp.

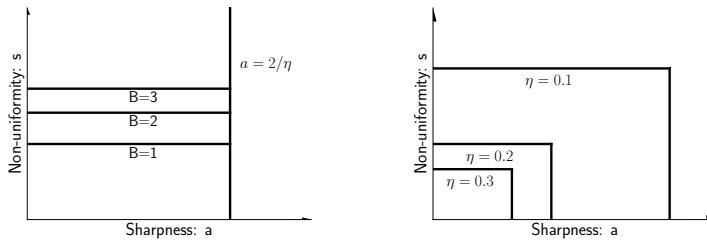

Figure 3: The sharpness-non-uniformity diagram, showing the rectangular region that is linearly stable for SGD. The left and right figure shows the influence of batch size $B$ and learning rate $\eta$, respectively. Notice that in the left plot, the stablility region of GD is the unbounded region between $a = 0$ and $a = 2/\eta$.

**The sharpness-non-uniformity diagram of SGD** Now assume that the learning rate $\eta$ is fixed. We use $a$ and $s$ as features of the global minima, to show how GD and SGD "selects" minima. From the results above, we know that the global minima that GD can converge to satisfy $a \leq 2/\eta$, and the global minima that SGD can converge to satisfy a more restrictive condition (9). The stability regions are visualized in Figure 3, which is called the sharpness-non-uniformity diagram.

From the sharpness-non-uniformity diagram we see that, when the learning rate is fixed, the set of global minima that are linearly stable for SGD is much smaller than that for GD. This means that compared to GD, SGD can filter out global minima with large non-uniformity.

### 3.2 Some Remarks

**Roles of the learning rate and batch size**    Our analysis shows that the learning rate and the batch size play different roles in global minimum selection. As shown in Figure 3, increasing the learning rate forces the SGD to choose global minima closer to the origin in the sharpness-non-uniformity diagram, which means smaller sharpness and smaller non-uniformity. On the other hand, decreasing batch size only forces SGD to choose global minima with smaller non-uniformity.

**Local stability for general loss functions**    As is well-known in the theory of dynamical system [11], the issue of asymptotic convergence to a particular critical point is only related to the local stability of that critical point, and locally, one can always make the linear approximation for the dynamical system or quadratic approximation for the objective function to be optimized, as long as the the linear or quadratic approximations are non-degenerate. Therefore our findings are of general relevance, even for problems for which the loss function is non-convex, as long as the non-degeneracy holds. However, as shown in Figure 4, the quadratic approximation, i.e. linearization of SGD, is not suited for the loss function shown in the solid curve, for which the Hessian vanishes at the minima. This happens to be the case for classification problems with the cross entropy used as the loss function. In this paper, we will focus on the case when the quadratic approximation is locally valid. Therefore in the following experiments, we use the mean squared error rather than the cross entropy as the loss function.

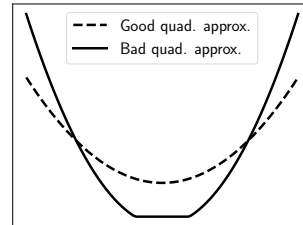

Figure 4

## 4    Experiments

In this section, we present numerical results[1] in deep learning in connection with the analysis above. We consider two classification problems in our experiment, described in Table 1. Since the computation of non-uniformity is prohibitively expensive, to speed it up, in most cases we only select 1000 training examples to train the models. For CIFAR10, only examples from the categories "airplane" and "automobile" are considered. We refer to Appendix C for the network architecture and computational method of sharpness and non-uniformity.

Table 1: Experimental setup

| Network type | # of parameters | Dataset | # of training examples |
|---|---|---|---|
| FNN | 898,510 | FashionMNIST | 1000 |
| VGG | 71,410 | CIFAR10 | 1000 |

### 4.1    Learning rate is crucial for the sharpness of the selected minima

First we study how the learning rate affects the sharpness of the solutions. We focus on GD but as we will show later, the general trend is the same for SGD. We trained the models with different learning rates, and report results on their sharpness in Table 2. All the models are trained for sufficient number of iterations to achieve a training loss smaller than $10^{-4}$. As we can see, by comparing the second

Table 2: Sharpness of the solutions found by GD with different learning rates. Each experiment is repeated for 5 times with independent random initialization. Here we report the mean and standard deviation of the sharpness in the second and third row of the table. The fourth row shows the largest possible sharpness predicted by our theory. Dashes indicate that GD blows up with that learning rate. Notice that GD tends to select the sharpest possible minima.

| $\eta$ | 0.01 | 0.05 | 0.1 | 0.5 | 1 | 5 |
|---|---|---|---|---|---|---|
| FashionMNIST | $53.5 \pm 4.3$ | $39.3 \pm 0.5$ | $19.6 \pm 0.15$ | $3.9 \pm 0.0$ | $1.9 \pm 0.0$ | $0.4 \pm 0.0$ |
| CIFAR10 | $198.9 \pm 0.6$ | $39.8 \pm 0.2$ | $19.8 \pm 0.1$ | $3.6 \pm 0.4$ | - | - |
| prediction $2/\eta$ | 200 | 40 | 20 | 4 | 2 | 0.4 |

and third rows with the fourth row, the numerical results are very close to the theoretical prediction

of the largest possible sharpness $2/\eta$, especially in the large learning rate regime. This may seem somewhat surprising, since the stability analysis only requires that the sharpness not exceeding $2/\eta$. Although there are lots of flatter minima, for instance those found by using larger learning rates, GD with a small learning rate does not find them. One tempting explanation for this phenomenon is that the density of sharp minima is much larger than the density of flat minima. Hence, when an optimizer is stable for both sharp and flat minima, it tends to find the sharp ones. It should also be remarked that the same phenomenon is not expected to hold for very small learning rates, since there are not that many really sharp global minima either.

## 4.2 SGD converges to solutions with low non-uniformity

Our theory reveals that a crucial difference between GD and SGD is that SGD must converge to solutions that fit all the data uniformly well. To verify this argument, we trained a large number of models with different learning rates and batch sizes, and show the results in Figure 5.

In Figure 5a, we plot the non-uniformity against batch size, where batch size $B = 1000$ represents GD. We see that the solutions found by SGD indeed have much lower non-uniformity than those found by GD. For example, in the CIFAR10 experiment, with a fixed learning rate $\eta = 0.01$, the non-uniformity of GD solutions is about $350$, however the same quantity is only about $100$ for SGD with batch size $B = 4$. This value is around half of the highest possible non-uniformity predicted by our theory, which is about $\sqrt{4}/\eta = 200$. Also we observe that the non-uniformity almost drops monotonically as we decrease the batch size. These suggest that our prediction about the non-uniformity is correct although not as tight as for the sharpness.

Figure 5b shows the influence of the batch size on the sharpness. We see that SGD always favors flatter minima than GD, and the smaller the batch size, the flatter the solutions. This phenomenon can not be explained directly by our theory. A possible explanation is provided in the next section.

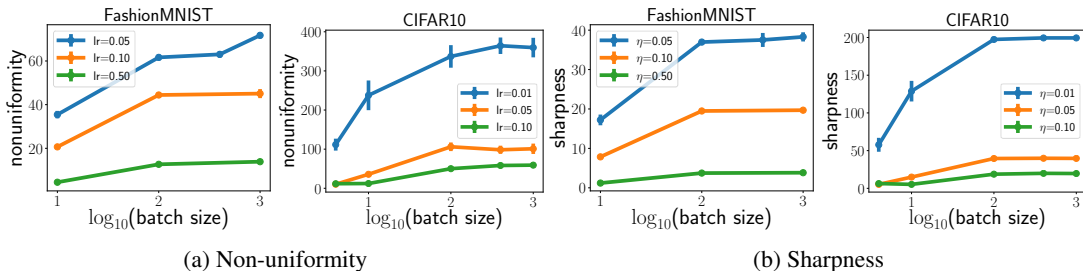

(a) Non-uniformity             (b) Sharpness

Figure 5: The influence of the batch size on the non-uniformity and sharpness. For each set of hyper-parameters, we trained the models with 5 independent random initialization and display the mean and standard deviation. The total number of samples is 1000, so the right-most values in each panel correspond to GD.

## 4.3 The selection mechanism of SGD

To investigate the accuracy of global minima selection criteria introduced in Section 3, we trained a large number of models, and display the results of the sharpness and non-uniformity in Figure 6. To take into account the influence of initialization, we tried three different initializations: uniform initialization $U[-v/\sqrt{n_{\mathrm{in}}}, v/\sqrt{n_{\mathrm{in}}}]$, with $v = 1, 2, 3$ for FashionMNIST and $v = 0.5, 1, 1.5$ for CIFAR10. Larger $v$ will cause the optimizers to diverge. We choose learning rate $\eta = 0.5$ for FashionMNIST and $\eta = 0.01$ for CIFAR10. For each specific set of hyper-parameters, we trained 5 model with independent initializations. The predicted largest possible non-uniformities by the theory are displayed by the horizontal dash lines.

As we can see, all the solutions lie within the area predicted by the theory. Specifically, for relatively large batch size, for example $B = 25$, the non-uniformity found for the solutions is quite close to the predicted upper bound. However, for very small batch size, say $B = 4$, the non-uniformities for both datasets are significantly lower than the predicted upper bound.

Another interesting observation is that the non-uniformity and sharpness seems to be strongly correlated. This partially explains why SGD tends to select flatter minima shown in Figure 5b, instead of sharp minima with low non-uniformity. As long as the non-uniformity is reduced, the sharpness is reduced simultaneously. This mechanism is clearly shown in Figure 6.

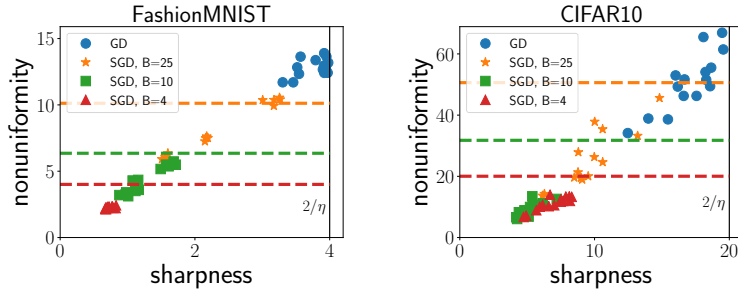

Figure 6: The sharpness-non-uniformity diagram for the minima selected by SGD. Different colors correspond to different set of hyper-parameters. The dash line shows the predicted upper bound for the non-uniformity. One can see that the data with different colors lies below the corresponding dash line.

The strong correlation between sharpness and non-uniformity is not part of the consequence of our theory. To further explore of the generality of this observed correlation,

we trained many more models with a variety of different learning rates, batch sizes and initialization. In addition, we considered a large-scale case, 14-layer ResNet for CIFAR-10 with $10,000$ training examples. We plot sharpness against non-uniformity in Figure 7. The results are consistent with Figure 6.

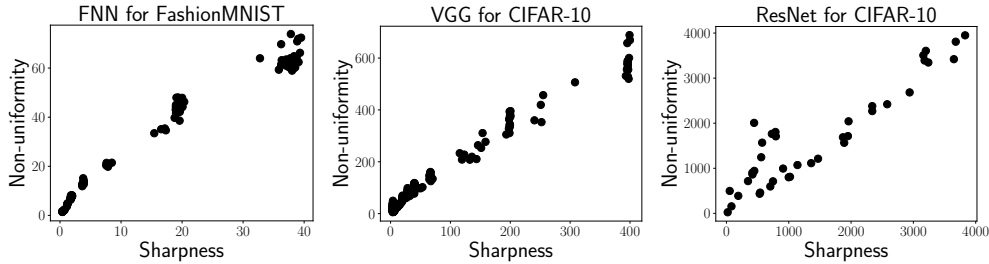

Figure 7: Scatter plot of sharpness and non-uniformity, suggesting that the non-uniformity and sharpness are roughly proportional to each other for these models.

## 4.4 Back to the escape phenomenon

Now we are ready to look at the escape phenomenon more closely. We first trained two models using GD with $\eta = 0.1$ to fit both the FashionMNIST and a corrupted FashionMNIST dataset. The latter contains an extra 200 training examples with random labels. The corrupted FashionMNIST is used as a more complex dataset, where the effect of the regularization is more significant. The information for the two solutions are summarized in Table 3. Starting from these two solutions, several SGD and GD with larger learning rates were run. Their dynamics are visualized in Figure 8.

Table 3: Information for the initializations of the escape experiment.

| dataset | test acc | sharpness | non-uniformity |
|---|---|---|---|
| FashionMNIST | 80.04 | 19.7 | 45.2 |
| Corrupted FashionMNIST | 71.44 | 19.9 | 51.7 |

**Prediction of escape** According to Table 3, the sharpnesses of both starting points are larger than 19.0. Our theory predicts that they are not stable for GD with learning rate larger than $2/1.9$. Therefore it is not surprising that GD with learning rate $0.3, 0.5$ will escape. For SGD with $\eta = 0.1, B = 1$ and $\eta = 0.1, B = 4$, the upper bound for the non-uniformity for being able to stay at the minima is at most $\sqrt{1}/0.1 = 10$ and $\sqrt{4}/0.1 = 20$. However, the non-uniformities of the two starting minima ($45.2$ and $51.7$) are too large. Hence, they are unstable for SGD with $\eta = 0.1, B = 1$ and $\eta = 0.1, B = 4$. In Figure 8, we see that all the previous predictions are also confirmed. In the corrupted FashionMNIST experiments, we also notice that SGD with $\eta = 0.1, B = 100$ fails to escape from the starting point. This is due to the fact that the non-uniformity of that solution ($51.7$) is smaller than $\sqrt{100}/0.1 = 100$. SGD with $\eta = 0.1, B = 100$ will be stable around that

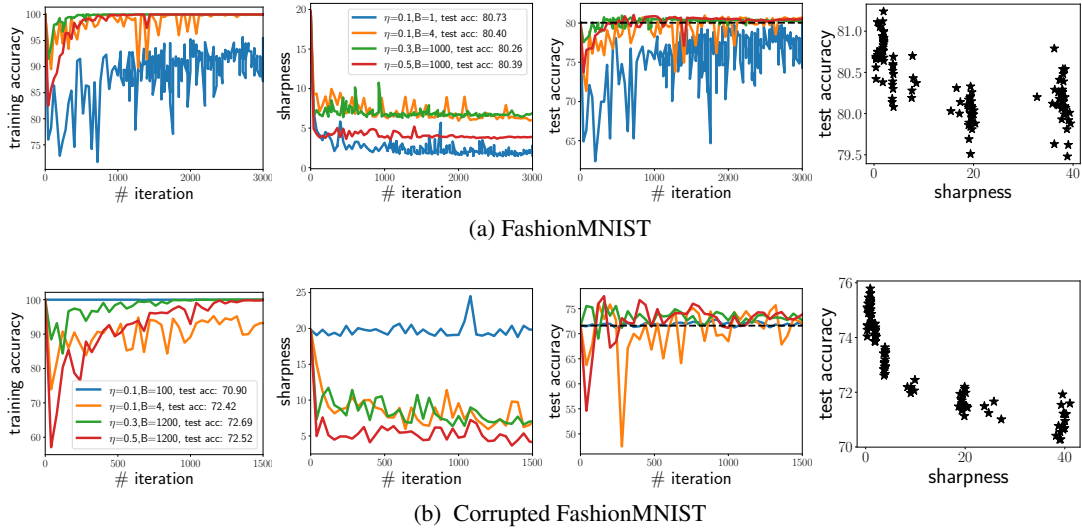

(a) FashionMNIST

(b) Corrupted FashionMNIST

Figure 8: The first three columns display the dynamics of the training accuracy, sharpness and test accuracy, respectively. To better show the escape process, we only show the first $1,500$ iterations. All the optimizers are run for enough iterations to achieve a training error smaller than $10^{-4}$, and the test accuracies of the final solutions are shown in the legends. The fourth column displays the scatter plots of the sharpnesses and the test accuracy of 200 models, which are obtained by using different learning rates, batch sizes and initializations.

point. Overall, we see that the escape can be predicted very well by using sharpness along with non-uniformity.

**The process of escape**    Now we focus on the first and second columns of Figure 8, i.e. the dynamics of the training accuracy and sharpness. We see that the dynamics display a sudden escape to some flatter area. After that the training error is gradually reduced, but the sharpness does not change much.

Also we can see that the escape process takes only a few iterations, much less than the number of iterations needed for the optimizers to converge (see the first column of Figure 8). This justifies our premise that linear stability is the real "driving force" for this escape phenomenon. Viewing SGD as SDE [8, 15, 6, 5] cannot explain this, since noise-driven escape is exponentially slow [1].

**Implications for generalization**    Let us examine the third column of Figure 8, together with the legends where the test accuracy of the final solutions is reported. As expected, GD with large learning rates and SGD with small batch sizes indeed converge to solutions that generalize better. Moreover, this effect is more significant for the complex dataset, corrupted FashionMNIST. This is expected from previous observations that sharpness and the generalization error are correlated [3, 7, 6]. To see the extent of this correlation, we plot the test accuracy against the sharpness in the fourth column of Figure 8. As we can see, the correlation between the test accuracy and the sharpness is stronger for the corrupted FashionMNIST dataset than for the clean FashionMNIST.

It should also be noted that one can construct examples in which flatter solutions have worse test accuracy. One such counterexample is given in Appendix B

# 5   Related Work and Discussion

The phenomenon of escape from sharp minima was first studied by Hu et al. [5], which suggested that escape from sharp minimizers is easier than for flat minimizers. Zhu et al. [15] suggested that the non-isotropic structure of SGD noise is essential for the fast escape. Jastrzębski et al. [6] suggested that the noise factor $\eta/B$ determines the sharpness of the solutions selected by SGD. A similar argument is used in Smith and Le [10] and Goyal et al. [2] in connection with test performance degradation in large batch training. These works viewed SGD as a diffusion process. In contrast, we analyzed SGD from a dynamic stability perspective. Our theory and experiments show that learning rate and batch size play different roles in the minima selection.

Yin et al. [14] proposed a quantity called gradient diversity defined by $\sum_{i=1}^{n} \|\nabla f_i\|^2 / \|\nabla f\|^2$ and used it to analyze the parallel efficiency of SGD. This quantity is similar in spirit to the ratio between non-uniformity and sharpness. However, it is not well-defined at the global minima for over-parameterized models, since $\nabla f_i = 0$ for any $i \in [n]$.

At a technical point, our work also bears some similarity with the convergence rate analysis of SGD in Ma et al. [9] since both focused on quadratic problem. However, it should be stressed that our interest is not at all the convergence rate, but whether it is possible for a particular optimization algorithm with a particular set of hyper-parameters to converge to a particular global minimum. Even though we also use formulas derived from quadratic problems to illustrate our findings, we should emphasize that the issue we are concerned here is local in nature, and locally one can almost always make the quadratic approximation. So we expect our results to hold for general (even non-convex) loss functions, whereas the explicit results of Ma et al. [9] should only hold for the quadratic problem they considered.

**Quasi-Newton methods**   Wilson et al. [12] found that compared to vanilla SGD, adaptive gradient methods tend to select solutions that generalizes worse. This phenomenon can be explained as follows. Consider the adaptive optimizer $x_{t+1} = x_t - \eta D_t^{-1} \nabla L(x_t)$, whose stability condition is $\lambda_{\max}(D^{-1}H) \le 2/\eta$. For algorithms that attempt to approximate Newton's method, we have $D \approx H$. Consequently almost all the minima can be selected as long as $\eta \le 2$. This suggests that these algorithms tend to select sharper minimizers. As an illustration, we used L-BFGS to train a model for corrupted FashionMNIST. It is observed that L-BFGS always selects minima that are relatively sharp, even though the learning rate is well tuned. This can be understood as follows. Starting from the best solutions selected by L-BFGS, both GD and SGD can escape and converge to flatter solutions which generalize better, as long as they are also well-tuned. We suspect that this might provide an explanation why adaptive gradient methods perform worse in terms of generalization. But further work is needed to see whether this really holds.

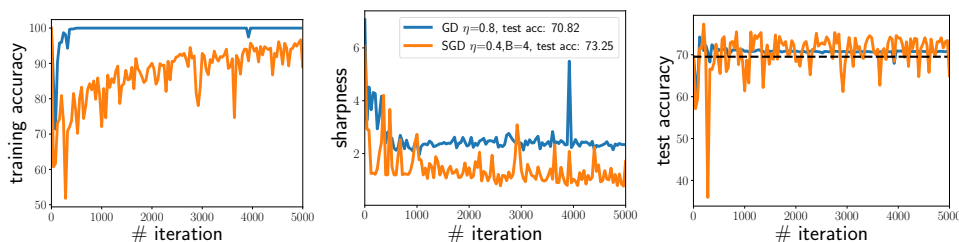

Figure 9: Escape of GD and SGD from the minima (test accuracy 69.5%) selected by well-tuned L-BFGS. The training accuracy, sharpness, and test accuracy with respect to the number of iterations are showed.

# 6   Conclusion

We have discussed the mechanism of global minima selection from the perspective of dynamic stability. Through the linear stability analysis, we have demonstrated that sharpness and non-uniformity are both important for the selection. For both GD and SGD, larger learning rates give rise to flatter solutions, and SGD tends to select minima with smaller non-uniformity than GD. For neural networks, it was observed empirically that non-uniformity is roughly proportional to the sharpness. This might explain why SGD tends to select flatter minima than GD in deep learning.

Regarding the connection to generalization, our current understanding is that it can go both ways. On one hand, one can construct examples for which sharper solutions generalize better. One such example is given in the appendix. On the other hand, there is plenty of evidence that flatness is strongly (positively) correlated with generalization error for neural networks. This work suggests the following picture for the optimization process. While GD can readily converge to a global minimum near the initialization point, SGD has to work harder to find one that it can converge to. During the process SGD manages to find a set of parameters that can fit the data more uniformly, and this increased uniformity also helps to improve the ability for the model to fit other data, thereby increasing the test accuracy. In any case, this is still very much a problem for further investigation.

**Acknowledgement**

We are grateful to Zhanxing Zhu for very helpful discussions. The worked performed here is supported in part by ONR grant N00014-13-1-0338 and the Major Program of NNSFC under grant 91130005.

## Footnotes

[1]The code is available at `https://github.com/leiwu1990/sgd.stability`

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
