[Supplementary Material · sgd_stability_supp.pdf]

## A Proof of Theorem 1

Without loss of generality, assume $x^* = 0$. Then the linearized SGD is given by

$$x_{t+1} = x_t - \frac{\eta}{B} \sum_{j=1}^{B} H_{\xi_j} x_t.$$

Hence, we have

$$\mathbb{E}x_{t+1} = \mathbb{E}(I - \eta H)x_t,$$

and

$$
\begin{aligned}
\mathbb{E}\|x_{t+1}\|^2 &= \mathbb{E}x_t^T \left[ I - \frac{2\eta}{B} \sum_{j=1}^{B} H_{\xi_j} + \frac{\eta^2}{B^2} \left( \sum_{j=1}^{B} H_{\xi_j} \right)^2 \right] x_t \\
&= \mathbb{E}x_t^T \mathbb{E} \left[ I - \frac{2\eta}{B} \sum_{j=1}^{B} H_{\xi_j} + \frac{\eta^2}{B^2} \left( \sum_{j=1}^{B} H_{\xi_j} \right)^2 \right] x_t \\
&= \mathbb{E}x_t^T \left[ I - 2\eta H + \frac{\eta^2}{B^2} \left( \frac{B(n-B)}{n(n-1)} \sum_{i=1}^{n} H_i^2 + \frac{nB(B-1)}{n-1} H^2 \right) \right] x_t \\
&= \mathbb{E}x_t^T \left[ (I - \eta H)^2 + \frac{\eta^2(n-B)}{B(n-1)} \Sigma \right] x_t.
\end{aligned}
$$

Therefore, if we have

$$\lambda_{\max} \left[ (I - \eta H)^2 + \frac{\eta^2(n-B)}{B(n-1)} \Sigma \right] \leq 1,$$

we have as $t \to +\infty$,

$$\mathbb{E}\|x_t\|^2 \leq \lambda_{\max}^t \left[ (I - \eta H)^2 + \frac{\eta^2(n-B)}{B(n-1)} \Sigma \right] \mathbb{E}[\|x_0\|^2] \leq \mathbb{E}[\|x_0\|^2],$$

therefore $x^*$ is linear stable.

If $d = 1$, then $H$ and $\Sigma$ are scalars, and we have

$$
\begin{aligned}
\mathbb{E}x_{t+1}^2 &= \left[ (1 - \eta H)^2 + \frac{\eta^2(n-B)}{B(n-1)} \Sigma \right] \mathbb{E}x_t^2 \\
&= \left[ (1 - \eta H)^2 + \frac{\eta^2(n-B)}{B(n-1)} \Sigma \right]^{t+1} \mathbb{E}[x_0^2]
\end{aligned}
$$

In this case, if

$$\left[ (1 - \eta H)^2 + \frac{\eta^2(n-B)}{B(n-1)} \Sigma \right] > 1,$$

then $\mathbb{E}x_t^2 \to \infty$, and $x^* = 0$ is not stable.

## B A Synthetic Example

In this section, we provide an example for which SGD selects solutions that generalize worse than GD.

In this example, the ground truth is $f^*(x) = 0$. We are given two data points $\{(0,0),(1,0)\}$ and we attempt to fit them using a second order polynomial parameterized by $f(x) := a_0 + \sqrt{a_1}x - \sqrt{a_2}x^2$. Thus the empirical risk is given by

$$J(a_0, a_1, a_2) := \frac{1}{2} \left( a_0^2 + (a_0 + \sqrt{a_1} - \sqrt{a_2})^2 \right), \tag{10}$$

with $a_1 \geq 0, a_2 \geq 0$. Here the global minima forms a one-dimensional manifold $S = \{(0, a, a) \mid a \geq 0\}$. Since the ground truth is $f^*(x) = 0$, models with a smaller value of $a$ generalizes better. For the global minima, we have

$$H_1 = \begin{pmatrix} 1 & 0 & 0 \\ 0 & 0 & 0 \\ 0 & 0 & 0 \end{pmatrix} \quad H_2 = \begin{pmatrix} 1 \\ \frac{1}{2\sqrt{a}} \\ -\frac{1}{2\sqrt{a}} \end{pmatrix} \begin{pmatrix} 1 & \frac{1}{2\sqrt{a}} & -\frac{1}{2\sqrt{a}} \end{pmatrix}^T.$$

The positive eigenvalues of the two matrices are $1$ and $1 + \frac{1}{2a}$, respectively. When $a$ is very small, they are very different from each other, i.e. the non-uniformity is large. They are close for large values of $a$. According to our analysis, SGD favors the area where $H_2 \approx H_1$. This means that SGD prefers solutions with larger values of $a$ than GD.

(a)          (b)          (c)

Figure 10: Fitting two points $\{(0,0), (1,0)\}$ with a second order polynomial. Both SGD and GD are initialized from $(a_0, a_1, a_2) = (0, 0.1, 0.2)$ with learning rate $\eta = 0.5$. **(a)** The trajectories of GD and SGD. **(b)** The solutions found by GD and SGD. **(c)** The histogram of $a_1 + a_2$ of the solutions found by GD and SGD, by running the optimizers 100 times.

We run both GD and SGD starting from $(0, 0.1, 0.2)$ with learning rate $0.5$ for $500$ steps. Figure 10 shows the results. In Figure 10a, we show the trajectory of GD and a realization of SGD. As we can see, SGD is unstable in the area near the initialization. It suddenly jumps to another area where $a$ is larger, and converges gradually to a minimum with large $a$. In contrast, GD is stable in the initialization area. It converges to a minimum close to the starting point (small $a$) without any jump. Since SGD has randomness, we ran this experiment for $100$ times, and report the histogram of $a_1 + a_2 = 2a$ of the converges results in Figure 10c. It clearly shows that with high probability SGD picks up solutions farther from the ground truth than GD.

## C    Details of Experiments

**Model Architecture**

- **FNN** A 4-layer fully connected network, which is used to fit FashionMNIST. The architecture is $784 - 500 - 500 - 500 - 10$.
- **VGG** A VGG-style network with 8 convolutional layers, which is used to fit CIFAR-10. The architecture is given in Table 4.
- **ResNet** A standard residual network with 14-layer convolutional layers. This network is used to fit CIFAR-10.

**Computation of the sharpness and non-uniformity** In this paper, the largest eigenvalues of both the Hessian matrix and the variance matrix are calculated by the power iteration, which is given as follows

$$v^k \leftarrow v^k / \|v^k\|$$
$$v_{k+1} = A v_k$$
$$\lambda_{k+1} = v_{k+1}^T v_k$$

Table 4: VGG network for CIFAR-10.

| Layer | Output size |
|---|---|
| input | $32 \times 32 \times 3$ |
| $3 \times 3 \times 16$, conv | $32 \times 32 \times 16$ |
| $2 \times 2$, maxpool | $16 \times 16 \times 16$ |
| $3 \times 3 \times 16$, conv | $16 \times 16 \times 16$ |
| $2 \times 2$, maxpool | $8 \times 8 \times 16$ |
| $3 \times 3\times$, conv | $8 \times 8 \times 32$ |
| $2 \times 2$, maxpool | $4 \times 4 \times 32$ |
| $3 \times 3 \times 64$, conv | $4 \times 4 \times 64$ |
| $2 \times 2$, maxpool | $2 \times 2 \times 64$ |
| $3 \times 3 \times 64$, conv | $2 \times 2 \times 64$ |
| $2 \times 2$, maxpool | $1 \times 1 \times 64$ |
| $64 \to 128$, linear | 128 |
| $128 \to 2$, linear | 2 |

where $v_0$ is sampled from $\mathcal{N}(0, I)$. In the experiments, we found that power iteration always converge within tens of iterations.

The matrix vector product $Av_k$ is computed by using the auto-differential functionality provided by PyTorch. For the Hessian matrix

$$H(x)v = \nabla(v^T \nabla f(x)).$$

For the covariance matrix $\Sigma = \frac{1}{n} \sum_i H_i^2 - H^2$, $H_i^2 v$ can be computed by

$$w = \nabla(v^T \nabla f_i)$$
$$H_i^2 v = \nabla(w^T \nabla f_i).$$

The above operation needs to construct a new computation graph for each sample, so the computation of non-uniformity is very expensive.