[Reviews · NeurIPS 2018]

Reviewer 1



This paper examines, theoretically and empirically, how does SGD select the global minima it converges to. It first defines two properties ("sharpness" and "non-uniformity") of a fixed point, and how these determine, together with batch size, the maximal learning rate in which the fixed point is stable under SGD dynamics (both in mean and variance). It is then demonstrated numerically how these results relate affect the learning rate and batch size affect the selection of minima, and the dynamics of "escape" from sharp minima". Clarity: This paper is nicely written, and quite clear. Quality: Seems correct, except some fixable errors (see below), and the numerical results seem reasonably convincing. Originality: The results are novel to the best of my knowledge. Significance: The results shed light on the connections between sharpness, learning rate, batch size, and highlight the importance of "non-uniformity". These connections are not well understood and have received attention since the recent work by Keskar et al. From the empirical side, I especially found it interesting that GD selects solutions with maximal sharpness and the strong correlation between sharpness and non-uniformity. I also agree with the authors that the fast escape from sharp minima (e.g., Figure 7, left column) favors this paper's "instability" explanation over the SDE interpretations (e.g., Jastrzebski et al.). Main Comments: 1) The linearization x_{t+1}=x_t - \eta A_{\xi_t}x_t is wrong: the zeroth order term G(0;\xi) is not necessarily zero (x^* is only a fixed point of G(x), not G(x;\xi)). In other words we need to keep constant term G(0;\xi) in the dynamics. These constants do not affect the dynamics of E[x_t] (since their mean is zero), but they do affect E[x^2_t] . This implies that the variance generally does not go to zero. An indeed, SGD generally doesn't converge to a fixed point without decreasing the learning rate to zero. A way to fix this would be to assume we converge to a global minima where \nabla G(x;\xi) for all \xi . This seems reasonable in overparameterized neural networks. 2) The numerical results focus on quadratic loss, since the authors claim that the Hessian vanishes for cross-entropy. This indeed happens, but only for minima at infinity. However, if we use cross-entropy with L2 regularization (as is commonly done), then all minima are finite and the Hessian would not vanish. Therefore, I'm not sure I understand this argument. Do the reported results also hold for cross-entropy with L2 regularization? Minor Comments: 1) I think eqs. 13 and 14 would be more helpful before eq. 8 (with A_i=H_i) than in their current location. 2) It would be helpful to see the derivation of eq. 6 in the appendix. 3) Figure 3 could use larger fonts (especially in labels) 4) Figure 6(a): "2/\eta" should not be there, right? 5) I recommend the authors to make their code available online to enable other to easily reproduce their results. Some Typos: line 217 - "the extend" line 224 - "is easily" line 292 - "generlizes" equation below line 314 - equality should be \leftarrow (since both sides have v^k). %% After author feedback %% I thank the authors for their reply. Some futher comments to improve the quality of the paper: First, I suggest the authors add complete proofs (in the SM), and the main Theorem they mention in the rebuttal. Second, please include citation and discussion related to the works mentioned by Reviewer 3. For example, with regard to Ma et al., which seems most relevant (the results in the first two papers assume strong convexity, which makes them generally inapplicable to over-parameterized models), I think the main differences are: 1) Ma et al. focused on how to select the learning rate and batch size to achieve an optimal convergence rate. In contrast, this paper focuses on how the batch size and learning rate determine when do we get dynamical instability and SGD escapes away from the minimum. 2) I think the resulting expressions are rather different. For example, the bounds in Ma et al. (e.g. Theorem 2) depend on the eigenvalues of the Hessian, while in this paper they depend on the maximal eigenvalues of H and Sigma. The latter expressions expose nicely the important factors affecting stability (sharpness and non-uniformity). Also, (and this is more minor) in Ma et al. the derived upper bound is for the expected loss, while here the bound is the first two moments of the optimized parameters. 3) I believe the results in this paper are more easily generalizable (to realistic models) since we only care about dynamical stability near the fixed point, and not the optimal learning rate/batch size (Which depend on the entire optimization path). 4) The derivation in this paper is quite simple and intuitive (at least for me). Third, the definitions of non-uniformity vary somewhat in the paper. Specifically: a. line 24 (intro): "the variance of the Hessian (see below for details)" b. line 66 (1D case): non-uniformity = s = std of the (1D) Hessian c. lines 90-91 (General case, matrix version): \Sigma (variance of Hessian matrix) is a "measure of the non-uniformity" d. lines 97-98 (General case, scalar version): non-uniformity = s, where s^2 is the largest eigenvalue of \Sigma e. Appendix C (what they used in practice): same as the previous definition (up to the approximation accuracy of the power iteration method). So definitions (b),(d), and (e) are consistent, and I think it is pretty clear that this what they actually mean by "non-uniformity". (c) is different, but they only say this is a "measure of .." and not the definition. I think this matrix version was mainly pointed out to give some intuition. (a) indeed seems to be inaccurate, and should be corrected, but I think it is a small mistake. I think these issues should be corrected, and a formal definition should be given, as non-uniformity is one of the key ingredients of this paper. Fourth, I think the authors should try to emphasize and clarify more the important parts in their experiments. I'll try to summarize the interesting observations I found in the experiments: 1) The sharpness value is nearly always near the maximum value (2/eta) for GD (Table 2, Figure 6a); Non-uniformity also has values with a similar magnitude as the bound (Figure 6a). 2) The linear relation between sharpness and non-uniformity (Figure 6) 3) Observations (1-2) above (as well as figure 3), suggest that smaller mini-batches decrease sharpness due to the non-uniformity bound (Figure 6). 4) Non-uniformity seems to predict well when we escape from sharp minima (Figure 7). 5) The escape from sharp minima is fast (Figure 7). I think this is quite a lot of new interesting information that I've learned here. Previous high impact papers suggested that * Keskar et al.: (a) "large batch size implies large sharpness" (b) "small sharpness is important for generalization" * Hoffer et al., Goyal et al., Smith and Le: (c) "changing the learning rate at large batch sizes can improve generalization" However, so far it was not clear why these statements should be true (e.g. counterexample in Appendix A for the statement b). I think the observations (1-4) above make significant progress in understanding statements (a-c), and suggests we should also focus on non-uniformity (instead of just sharpness). Also, observation 5 suggests that previous SDE models for this phenomena are wrong. Other previous explanations (e.g., Yin et al.) seems not be applicable to over-parameterized case, as explained in the discussion. In terms of math, I am not aware of other works that examined this specific over-parameterized case, where all sample gradient are zero at the minimum, except for Ma et al. (which focused on different questions). Fifth, I think that there is an issue in the conditions relying only on the maximal eigenvalues. For example, in at line 98 the author state that "a sufficient condition for (9) to hold is eq. (10)", and condition (10) only depends on the maximum eigenvalue of H. However, I don't see why small (possibly even zero) eigenvalues of H can't affect stability. For example, suppose that B=1 and \eta=1, that v is the eigenvector corresponding the max eigenvalue of (9), and also that H*v=c*v, \Sigma*v = q*v for two positive scalars c and q. In this case, we get an eigenvalue of the matrix in (9) which is equal to (1-c)^2+q. This value can be larger than 1, if c is sufficiently small (e.g. c=0), relatively to q. The authors should explain how this issue could be avoided, or correct their statement. In any case, I recommend adding more mathematical details for all calculations in the supplementary.

Reviewer 2



============ after rebuttal Some of my concerns in my original review have been addressed by the authors, and would like to commend them for following up with more experiments in the rebuttal (props to the new format). I overall think that this is a solid and very interesting paper, and will upgrade my score from 6 to 7. ============ original review The authors aim to connect sharpness and variance of a given solution found by SGD with its generalization ability. The results presented are motivated from simple quadratic problems, and then are validated on experimental setups. This is a mostly experimental paper on the sources of model generalization, and uses simple quadratic models to motivate the presented conjectures. I found the results of this paper extremely interesting. I also found the simplified examples quite interesting and intuitive. For example, Fig2 is quite enlightening in all its simplicity. I have not seen something similar, and although simple, it gives rise to an interesting observation. My main concerns are mostly on presentation and clarity, and also on the limited number of experimental data. Here are my comments on the paper: - the use of the word “stability” in this paper is very non-standard. Stability (in the context of generalization) is used to explain how robust are output models to perturbing the data set by a little bit (by say removing one sample), eg check the work by Elisseeff and Bousquet. - in Section 2.2 it’s unclear if the authors focus on the simple quadratic problem of (1), or something more general - I was a little confused by section 2.3 In this case there is only 1 global min. In this case, I don’t understand what the value of sharpness vs non-uniformity implies. - In sec 2.3 the authors mention “At the level of matrices, H is a measure of the sharpness and Sigma is a measure of the non-uniformity. This is hard to parse as matrices themselves are not “measures” I don’t really understand what the sentence is trying to explain. - It was not clear to me why the condition is (9) is needed for convergence. - The authors mention that “ In the old days when typical models have only few isolated minima, this issue was not so pressing.”. What does that even mean? Adding one extra parameter creates infinite many global minima. -“The task of optimization algorithms has become: Find the set of parameters with the smallest testing error among all the ones with no training error.” Not true, optimization algorithms still find the model that minimizes risk. - “one observes a fast escape from that global minimum and subsequent convergence to another global minimum.” Unclear if that is what is happening in Fig 1. At 2k iterations both train and test acc stabilize, and before that train acc was *not* at 100% - “the ultimate global minimum found by SGD generalizes better” Unclear as the SGD model flactuates around the performance of the GD model and is always within +- 5%. - I think the experiments are quite interesting, but also limited. For example, it is unclear how well sharpness and non-uniformity would correlate with gen error in CIFAR10. How does the behavior change when using Resnets or DenseNets Overall, this paper presents very interesting observations that I have not seen in other papers, however the presentation needs quite some more work, as the clarity of many of the sections is lacking in parts. typos: accessible by an stochastic gradient -> accessible by a stochastic gradient In other words, how the different optimization algorithms -> In other words, how do the different optimization algorithms though the GD iterations is reasonably -> though the GD model is reasonably We then extend this analysis to high dimensional case. -> We then extend this analysis to the high dimensional case and hence accessible for SGD -> and hence accessible by SGD The left two diagrams in Figure 3 shows these two domains -> The left two diagrams in Figure 3 show these two domains Dashes indicate that GD blows up with that learning rate. -> Dashes indicate that GD diverges with that learning rate.

Reviewer 3



This paper investigates the behavior of the convergence of SGD (with a fixed learning rate \eta and batch size B) to a global minimum in an over-parameterized ERM problem. The paper introduces two quantities called non-uniformity and sharpness (which capture the variance of the Hessian at the global minimum), and argues that the ability of SGD to reach a global minimum depends on these quantities. The paper focuses mainly on the quadratic case in the over-parameterized regime, and derives necessary conditions on the aforementioned quantities (in terms of the fixed step size \eta and batch size B) for SGD to converge to the global minimum (Sec 2.3). Except for the toy example, which I find interesting, the main claims in this paper seem to be a direct consequence of the previously known results/analyses concerning the convergence of SGD with fixed learning rate. There have been several works that studied and analyzed the convergence rate of SGD and derived selection roles for the fixed learning rate in various settings (e.g., [Moulines-Bach, NIPS 2011], [Needell et al., NIPS 2014], [Ma et al., ICML 2018]). The last reference analyzes the learning rate of mini-batch SGD as a function of the batch size and some properties of the Hessian in the over-parametrized regime . In fact, the conditions in Sec 2.3 seem to be related to the analysis for the quadratic loss in that reference. That is, it seems that the characterization of the "non-uniformity" and "sharpness" in terms of the learning rate and batch size in this paper is essentially equivalent to the problem of characterizing the learning rate in terms of the batch size and the properties of the loss, which has been studied in several previous works. So, I don't really see much of a technical novelty in the paper. That said, I think the paper started with an interesting toy example in an interesting setting (non-convex functions with multiple global minimizers), which seemed to be the "right" setting to study the phenomenon of "SGD favoring flatter minima". However, the discussion suddenly went back to the convex (quadratic) case for the remainder of the paper. It would be interesting if the claim is formally substantiated in a more general class of objective functions. ------------------------------ ------------------------------ **** Post Rebuttal Comments: All in all, after reading the authors' rebuttal and the views of other reviewers, I think, despite the technical similarity with previous works, the paper looks at the problem from a different angle and may provide a new perspective on a very important problem. For this, I would increase my initial score by 1 point. I do think nonetheless that the authors should include proper citations to the relevant works mentioned in the review and discuss the connections to/differences from these works so that the context and the message of the paper becomes more clear.